# Humoral and Cellular Response after mRNA Vaccination in Nursing Homes: Influence of Age and of History of COVID-19

**DOI:** 10.3390/vaccines10030383

**Published:** 2022-03-02

**Authors:** Jesús San Román, Francisco Javier Candel, Juan Carlos Sanz, Paloma López, Rocío Menéndez-Colino, Pablo Barreiro, María del Mar Carretero, Marta Pérez-Abeledo, José Manuel Viñuela-Prieto, Belén Ramos, Jesús Canora, Raquel Barba, Antonio Zapatero-Gaviria, Franciso Javier Martínez-Peromingo

**Affiliations:** 1Department of Medical Specialties and Public Health, Universidad Rey Juan Carlos, 28922 Madrid, Spain; jesus.sanroman@urjc.es; 2Unit of Microbiology, Regional Laboratory of Public Health, Hospital Enfermera Isabel Zendal, 28055 Madrid, Spain; franciscojavier.candel@salud.madrid.org (F.J.C.); juan.sanz@salud.madrid.org (J.C.S.); maria.carretero@salud.madrid.org (M.d.M.C.); marta.perez.abeledo@madrid.org (M.P.-A.); vinuelajm@hotmail.com (J.M.V.-P.); belen.ramos@salud.madrid.org (B.R.); 3Clinical Microbiology and Infectious Diseases, IdISSC and IML Health Institutes, Hospital Universitario Clínico San Carlos, 28040 Madrid, Spain; 4Nursing Homes and Long-Term Care Facilities Support Unit, Dirección Asistencial Norte, Gerencia Asistencial Atención Primaria, 28035 Madrid, Spain; plhernandez@salud.madrid.org; 5Service of Geriatrics, Hospital General Universitario La Paz, 28046 Madrid, Spain; rocio.menendez@salud.madrid.org; 6Department of Neurosurgery, Hospital General Universitario La Paz, 28046 Madrid, Spain; 7Service of Internal Medicine, Hospital Universitario de Fuenlabrada, Universidad Rey Juan Carlos, 28922 Madrid, Spain; jesus.canora@salud.madrid.org; 8Service of Internal Medicine, Hospital Universitario Rey Juan Carlos, Universidad Rey Juan Carlos, 28922 Madrid, Spain; raquel.barba@hospitalreyjuancarlos.es; 9Vicecounselor of Healthcare and Public Health, Universidad Rey Juan Carlos, 28922 Madrid, Spain; antonio.zapatero@salud.madrid.org; 10Service of Geriatrics, Hospital Universitario Rey Juan Carlos, Universidad Rey Juan Carlos, 28922 Madrid, Spain; fmperomingo@salud.madrid.org

**Keywords:** COVID-19, SARS-CoV-2, nursing homes, older adults, occupational, vaccine, BNT162b2, serology, interferon-gamma release assay

## Abstract

Background: Most residents and staff in nursing homes have received full vaccination. Factors related to the immune response to vaccination might be related to the risk of future severe COVID-19 and may guide the need for vaccine boosters. Design: Nursing homes that were tested in a point survey in July-October 2020 were again analyzed after a vaccination campaign in June-July 2021. Immune responses according to IgG against nucleocapsid and spike antigens, and CD4 and CD8 interferon-gamma release assay against spike antigens, were evaluated. Results: A total of 1973 subjects were tested (61.7% residents, 48.3% staff), with a mean (SD) follow-up of 46.4 (3.6) weeks between assessments. More than half of residents and more than a third of staff had evidence of COVID-19 before vaccination; 26.9% and 22.7% had seroreversion of IgG-N, and 8.9% and 4.6% had IgG-N seroconversion at second assessment, respectively. Up to 96.8% of residents and 98.1% of workers had positive IgG-S after a mean of 19.9 (2.1) weeks after vaccination. In residents with vs without a history of COVID-19, IgG-S titers were 4.11 (0.54) vs. 2.73 (0.74) logAU/mL (*p* < 0.001); in workers these titers were 3.89 (0.61) vs. 3.15 (0.64) logAU/mL (*p* < 0.001). Linear regression analysis showed that younger age (OR: −0.03 per 10 years-older [95% CI, −0.04 to −0.02], *p* < 0.001) and evidence of COVID-19 (OR: 1.14 [95% CI, 1.08 to 1.20], *p* < 0.001) are associated with greater IgG-S titers after vaccination. A direct association was found between IgG-S titers and the intensity of IFN-gamma response against spike antigens. Conclusions: Waning of humoral response and reinfection seems to be more frequent in older as compared to younger adults, although cellular responses shortly after vaccination are comparable between these groups. Younger age and prior COVID-19 are related to greater humoral response after vaccination against SARS-CoV-2.

## 1. Introduction

During the first months of the COVID-19 pandemic [1,2], nursing homes for older adults concentrated high levels of morbidity and mortality of residents and also high levels of infections among workers. People in these centers were prioritized for SARS-CoV-2 vaccination. According to recent observations, the benefits of vaccination against SARS-CoV-2 include, in order of hazard reduction, lowering mortality, frequency of severe disease, and incidence of infections [3].

The frequency of SARS-CoV-2 reinfections, or breakthrough infections of those vaccinated, are yet to be determined, but seem to be greater for older adults as compared with younger people [4]. Associated factors include the selection of new viral variants [5] and waning immunity [6,7].

For older adults [8], the immune enhancement from booster doses of vaccine reduces the rate of infections and of severe disease; but the question remains whether other populations require additional vaccination, or if other factors, such as prior COVID-19, may preclude this intervention [9]. Herein we have analyzed the humoral response after BNT162b2 vaccination in residents and workers in nursing homes, and we have compared the immune profile after vaccination between those with and without a history of COVID-19.

## 2. Methods

Nursing homes that were under follow-up by the Supporting Unit for Residencies (SUR) in the northern area of the Community of Madrid were offered the opportunity to participate in this study. All centers were assessed with a serological point-survey between July and October 2020 (time 1) [10]. For that study, all residents and workers in nursing homes were tested for their serological response to SARS-CoV-2 prior to the initiation of the vaccination campaign. For the present study only vaccinated individuals (vaccinated with the BNT162b2 vaccine (Comirnaty^®^, Pfizer-BioNTech^®^, Mainz, Germany) with the first dose given by the end of December 2020, or the beginning of January 2021, and a second dose given in the second half of January and February 2021) were selected (time 2). Participation in the study was voluntary and included written informed consent. The study was approved by the Ethics Committee of the Hospital Clínico San Carlos under the name ‘SeroVAC study’ (reference number 21/274-O_M_SP).

The qualitative detection of IgG against nucleocapsid (N) and the qualitative and semi-quantitative detection (in arbitrary units per mL [AU/mL]) determinations of IgG against the spike (S) antigens of SARS-CoV-2 were carried out by chemiluminescent microparticle immunoassays (CLIA) (Abbott^®^ Ireland Diagnostics Division, Sligo, Ireland) and the ARCHITECT System (Abbott^®^, Chicago, IL, USA). The cut-off of positivity for IgG-S was ≥50 AU/mL according to previous validation studies [11]. For subjects with IgG-S concentrations above the upper limit of the analytical measuring interval a (40,000 AU/mL) an arbitrary concentration of antibodies of 2-fold above this level was considered.

For a subset of participants, the interferon-gamma (IFNγ) release assay (IGRA) was performed using SARS-CoV-2 S peptide formulations antigen 1 (Ag1) and antigen 2 (Ag2) and stimulation in whole blood (QuantiFERON^®^ SARS-CoV-2). All residents and workers at a single center that was randomly selected were assessed for cellular immunity at time 2. The production of IFN-γ was measured using the Sandwich CLIA platform approved for the determination of cellular immunity against M.-tuberculosis-specific antigens (QuantiFERON-TB Gold Plus. LIAISON XL^®^, DiaSorin^®^, Saluggia, Italy) [12], but in this case mycobacterial reactants were substituted with SARS-CoV-2 antigens (QuantiFERON SARS-CoV-2 Research Use Only, Qiagen^®^. Hilden, Germany) for in vitro stimulation of lymphocytes. Venous whole blood samples were collected directly in a core tube with lithium heparin, and later transferred to the QuantiFERON 2 tubes containing S peptides (Ag1 and Ag2), as well as positive (mitogen) and negative (nil) controls. The Ag1 and Ag2 are currently in vitro diagnostic products labeled for Research Use Only (RUO) and are not yet validated for clinical purposes. Specimens were processed as per the manufacturer’s guidelines [13,14,15]. The CLIA platform determines IFN-γ concentrations in international units per liter (IU/L), although the recommendation is that for clinical purposes a qualitative result is produced using a cut-off titer that is yet to be determined for SARS-CoV-2 infection. To calculate the final results per patient, the nil control test needs to be subtracted from mitogen, Ag1, and Ag2 results. Final IFN- γ concentration in the mitogen control test needs to be >500 IU/L to validate the final Ag1 and Ag2 results. The cut-off of positivity in IGRA used (≥25 IU/L for Ag1 and Ag2) has been previously established at the Public Health Regional Laboratory of the Community of Madrid in a case-control pilot study [16]. 

Statistical analyses were done using SPSS^®^ software, version 20 (IBM^®^, Chicago, IL, USA). Student’s t test was used to compare normally distributed continuous variables. In the case of non-normally distributed variables, Mann-Whitney’s U test was applied. Comparison of proportions for categorical variables was done either by chi-square or Fisher’s exact tests, as required. Correlation tests were done using Spearman’s rho test. Linear regression analysis was done for the study of variables associated with IgG-S concentration after vaccination; key assumptions for this test were met (Durbin–Watson test, uniform distribution of studentized residual, Pearson’s correlation between variables, no collinearity and normal distribution of residuals).

## 3. Results

A total of 1973 subjects living as residents (n: 1218; mean (SD) age: 83.7 (12.1) years-old; 71.2% women) or working as staff members (n: 755; mean age: 47.0 (11.4) years-old; 85.3% women) in 44 nursing homes for older adults were tested at two time points, in July to October 2020 (time 1) and in June to July 2021 (time 2). Table 1 shows the primary characteristics of residents and workers included in the study. The mean lag between the two serological assessments (time 1 to time 2) was 46.4 (3.6) weeks.

At time 1, evidence of prior COVID-19 among residents was found in 539 (44.2%) subjects by positive IgG-N and in 608 (50.0%) by positive IgG-S (*p* = 0.05); as for staff, the results were 204 (27.1%) IgG-N positive, and 270 (35.9%) IgG-S positive (*p* < 0.001). A total of 619 (50.8%) residents and 277 (36.7%) staff members had evidence of COVID-19 before vaccination either due to being IgG-S and/or IgG-N positive (*p* < 0.001).

In the period between time 1 and time 2, with respect to SARS-CoV-2 IgG-N positivity in residents, 328 (26.9%) showed seroreversion (IgG-N positive to IgG-N negative) and 109 (8.9%) seroconversion (IgG-N plus IgG-S negative to IgG-N positive); in the case of workers, we found 171 (22.7%) seroreversions and 35 (4.6%) recent infections using the same criteria. The proportion of seroreversions (*p* = 0.003) and of recent infections (*p* < 0.001) was greater in residents as compared with staff.

At time 2, after all individuals had received two doses of BNT162b2 mRNA vaccine, 1180 (96.8%) residents and 740 (98.1%) workers had positive IgG-S (*p* = 0.09). Mean lag between the second dose of vaccine and postvaccination serology was 19.95 (2.06) weeks. At this time point, 323 (26.5%) residents and 70 staff members (9.3%) had positive IgG-N tests. Evidence of SARS-CoV-2 infection at any time, in the form of having positive IgG-N at time 1 or time 2, was found in 727 (59.7%) residents and in 311 (41.2%) workers at time 2 (*p* < 0.001). Geometric mean titers of IgG-S, in subjects with detectable IgG-S at time 1, were 1.65 (1.45) logAU/mL among residents and 1.19 (1.32) logAU/mL among workers (*p* < 0.001), and at time 2 in all vaccinated subjects they were 3.55 (0.93) logAU/mL among residents and 3.46 (0.73) logAU/mL among workers (*p* < 0.001). At time 1 there were greater IgG-S concentrations in subjects with a history of COVID-19. At time 2, after vaccination among residents, IgG-S titers were 4.11 (0.54) logAU/mL in subjects with a history of COVID-19 and 2.73 (0.74) logAU/mL in subjects with no history of COVID-19 (*p* < 0.001). In workers these IgG-S levels were 3.89 (0.61) logAU/mL and 3.15 (0.64) logAU/mL (*p* < 0.001), respectively. After vaccination, as compared with before vaccination, IgG-S concentrations always increased significantly both in residents and staff, and in those with or without a history of COVID-19 (*p* < 0.001) (Figure 1). In residents and staff with a history of COVID-19 at time 1, the mean increase in IgG-S titers was 4.04 (0.46) logAU/mL in residents and 3.99 (0.44) log AU/mL in staff, as achieved after vaccination at time 2 (*p* = 0.004).

In a subset of these subjects (28 residents and 15 staff) we were able to test cellular immunity at time 2 by IGRA after stimulation of peripheral blood lymphocytes with SARS-CoV-2 Ag1 and Ag2. A total of 24 (85.7%) residents and 14 (93.3%) workers (*p* = 0.5) were IGRA positive and 5 subjects were IGRA negative. Taking together 40 residents and staff in whom paired humoral and cellular immunity tests were available, a direct association was found between IgG-S titers and the intensity of IFNγ response to Ag1 and Ag2 (Figure 2).

Linear regression analysis including age, sex, personal status (resident or staff), and serological evidence of COVID-19 (positive IgG-N) showed that IgG-S concentration after vaccination was associated with age (OR: −0.03 per 10 years-old (95% CI, −0.04 to −0.02 [*p* < 0.001]) and evidence of past COVID-19 (OR: 1.14 (95% CI, 1.08 to 1.20 [*p* < 0.001]) (Table 2).

## 4. Discussion

We observed that targeting the spike (S) protein is a better marker of SARS-CoV-2 infection than the nucleocapsid (N). Although both viral proteins are highly immunogenic, the decay in the levels of antibodies is faster for IgG-N than for IgG-S [17,18].

During the initial phases of the pandemic [10], in agreement with previous studies, our data confirm the high rate of infections in nursing homes. Seroreversion of IgG-N occurred for close to a quarter of subjects after approximately a year; yet even more remarkable was the difference between residents and staff [19]: the proportion of residents with seroconversion was almost twice that of staff—with the implication that the waning humoral immunity of older adults places them at greater risk for reinfection [20,21].

Older adults living in nursing homes, even though vaccinated, may still be at high risk for a complicated bout of COVID-19, given their age, frailty, and need for care. These breakthrough infections of older adults, but not members of the staff, have been previously reported [22]. Greater age is related to lower response to SARS-CoV-2 vaccination [9]. We observed, in the linear regression analysis, that IgG-S levels after vaccination are independently reduced with greater age. Other studies have shown a faster drop in the concentration of vaccine-induced immunity of older adults who are naïve versus those who are SARS-CoV-2-experienced [23]. Age, however, was not a factor affecting long-term antibody levels of patients convalescent with COVID-19 in another study [24]. As older patients may have more serious morbidity with reinfection or breakthrough infection [25], booster doses of vaccines are clearly indicated for residents of nursing homes. It is encouraging to observe in our study that the initial immune responses are in the short-term comparable among older and younger individuals.

Patients with a history of COVID-19 before vaccination have been shown to have greater IgG-S levels after receiving the vaccine as compared with those with no exposure to natural infection [9,26]. Further studies are required to determine whether subjects with a history of COVID-19 are better protected from infection or severe disease after vaccination, or if they would benefit from a different boosting protocol.

Evidence that T-cell immunity provides protection from viral replication, facilitates viral clearance, and sustains humoral response is consistent with our observation that most vaccinated subjects have positive IGRA to SARS-CoV-2 [27,28]. The small number of subjects tested means that these conclusions should be taken with caution. Antibody titers also may not accurately reflect the level of protection; according to a recent study [29], T-cell response shows little signs of exhaustion after vaccination. The correlation between humoral and cellular immunity, although statistically significant is still weak, may be in part due of the heterogeneity in the population, particularly with respect to age.

Some limitations of this study should be taken into consideration. We had no access to medical records and it is very likely that health status and comorbidities are factors affecting immune response to vaccination, both for residents and staff. Given the high risk of infection afflicting nursing homes, the set of residents included is biased as survivors were probably those with a better genetic background, clinical profile, and more robust immunity. Despite this selection bias, older adults are still at more risk for morbidity by SARS-CoV-2 than younger individuals. The design of the study allowed us to obtain clinical and laboratory information at two points, but we have no data about other factors within the times of assessment (i.e., other episodes of SARS-CoV-2 infection, other clinical events, treatment that may alter immune response, etc.). The selection of subjects in nursing homes was not properly performed after calculated sampling, making conclusions unable to be extrapolated. However, having found a similar proportion of seropositive residents and staff at time 1, a large point survey undertaken in the Community of Madrid [10] may indicate that the population studied was representative.

In conclusion, humoral immune waning seems to be more frequent in older adults. Antibody responses shortly after vaccination are less intense in older than in younger individuals. Further study could answer whether prior COVID-19 provides more durable immunity and lower risk upon reinfection with SARS-CoV-2 and, therefore, may make these subjects less vaccine-dependent.

## Figures and Tables

**Figure 1 vaccines-10-00383-f001:**
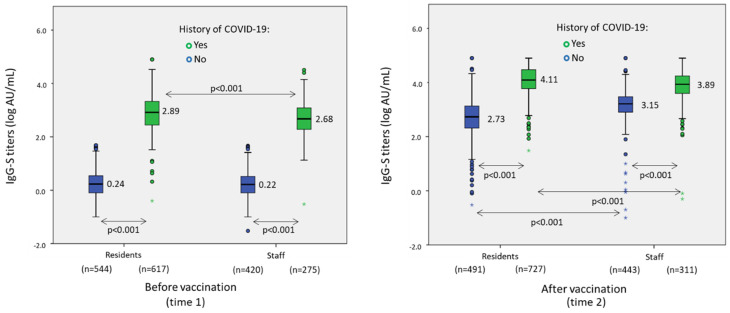
Association between IgG and spike titers before and after BNT162b2 mRNA-SARS-CoV-2 vaccination and history of COVID-19 (as defined by positive IgG-N and/or IgG-S), in residents and staff in nursing homes.

**Figure 2 vaccines-10-00383-f002:**
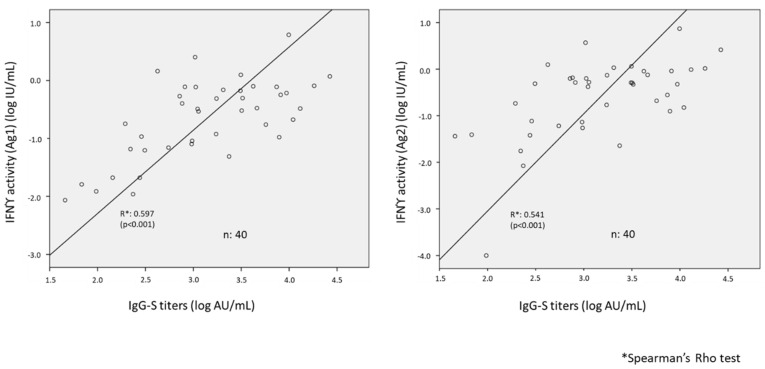
Association between humoral (IgG to spike titers) and cellular (IFNγ production) response after mRNA SARS-CoV-2 vaccination.

**Table 1 vaccines-10-00383-t001:** Main characteristics of the population of study.

	Residents	Workers	*p*-Value
Number (%)	1218	755	
Female sex (%)	867 (71.2)	644 (85.3)	
Mean age (SD) (years-old)	83.7 (12.1)	47.1 (11.4)	
Evidence of SARS-CoV-2 infection at time 1 (%)			
IgG-N positive	539 (44.2) ^+^	204 (27.1) *	<0.001
IgG-S positive	608 (50.0) ^+^	270 (35.9) *	<0.001
Any IgG positive	619 (50.8)	277 (36.7)	<0.001
Evidence of SARS-CoV-2 infection at time 2 (%)			
IgG-N positive	727 (59.7)	311 (41.2)	<0.001
Evolution of serology from time 1 to time 2			
IgG-N reversion	328 (26.9)	171 (22.7)	0.003
IgG-N conversion	109 (8.9)	35 (4.6)	<0.001
IgG-S positive	1180 (96.8)	740 (98.1)	0.09

N, nucleocapsid protein; S, spike protein; ^+^ *p* = 0.05; * *p* < 0.001.

**Table 2 vaccines-10-00383-t002:** Linear regression analysis of factors associated with IgG-S titers (logAU/mL) after vaccination against SARS-CoV-2.

	OR	95% CI	*p*
Age (per 10 years older)	−0.03	−0.04 to −0.02	<0.001
History of COVID-19 (yes vs. no)	1.14	1.08 to 1.20	<0.001
Female vs. male sex	−0.015	−0.08 to 0.05	0.67
Staff vs. residents	−0.01	−0.10 to 0.12	0.87

## Data Availability

The data presented in this study are available on request from the corresponding author. The data are not publicly available due to privacy issues.

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
