# Peer review of "Humoral and Cellular Response after mRNA Vaccination in Nursing Homes: Influence of Age and of History of COVID-19"

_vaccines, 2022, doi:10.3390/vaccines10030383_

Round 1

Reviewer 1 Report

The manuscript has been prepared correctly. The authors described a fairly large group of participants (1,972), but in fact they performed a complete set of studies on about 40 of them. Nursing home residents and their staff are one of the most frequently described groups of respondents (next to medical unit employees), due to the high risk of infections and their complications, rapid implementation of the vaccination program and a high degree of vaccination. Many more detailed scientific articles for this group have already been published (e.g. Cabezas et al. 2021, Canaday et al. 2021, Oyebanji et al. 2021, Tut et al. 2021, Van Praet et al. 2021, Míguez et al. 2022), therefore the results obtained do not provide significantly new information. The more so as the results concern people who received two doses of the vaccine, while in many countries this group has already received the third vaccination. The effect of previous COVID-19 on a significant increase in antibody levels is already well known. Conclusions regarding the influence of age on the level of antibodies are indirect and may be biased by the fact that the influence of other factors on the humoral response was not taken into account.

Besides, there are a few bugs and inaccuracies:

The methods should start with a description of the test group, along with a table containing the characteristics of the participants. They are described in the text, but most of them in the results, which is quite a non-standard arrangement. This method makes it difficult to search for basic information and interpret it.

When describing the IGRA test, it is necessary to take into account the relevant controls, and if so, what were their reference values. Describe the stimulation conditions or an annotate that all steps have been performed according to the manufacturer's protocol.

There is not description of the statistical methods used. Please, provide names of the tests performed.

There is not information whether the research was approved by the bioethical committee. The QuantiFERON® SARS-CoV-2 assay is licensed by the RUO, so it was not a routine diagnosis.

The results of the qualitative detection of antibodies against the nucleocapsid protein should be provided. The manuscript (line 62) mentions that such analysis has been performed. Whether it was performed on all participants and whether the test result was a criterion by which the convalescent status was confirmed.

On the basis of which criteria were selected people who underwent the IGRA test and how many actually were there should be provided. There is a large disproportion between the group where the antibody tests were performed (1,972) and the people who underwent the IGRA test (approx. 40). The authors wrote in the results that they tested 28 patients and 15 employees, and obtained a positive result for 24 patients and 14 employees (lines 118-121). Does this mean that some results were excluded? And if so, on what basis? It seems to me that with such a small representation of the results for the IGRA test, the conclusion regarding the general population is unfounded (line 121-123).

There were no detailed analyzes of the influence of age itself (the inference is rather indirect, based on the assumption that the charges are older than the staff). Age categories (e.g. in 10-year intervals) should be introduced in each group and the statistical significance of differences in antibody levels between age groups should be demonstrated. The health condition of the participants themselves (this characteristic is missing) and the existing diseases are probably not without significance, and they will rather affect the humoral response, not the age itself, in the compared groups.

Data (if available) should be included concerning reinfection or SARS-CoV-2 infection following full vaccination in these groups.

Author Response

The methods should start with a description of the test group, along with a table containing the characteristics of the participants. We have described the population selected for the study and we have included a table with the description of the subjects recruited.

When describing the IGRA test, it is necessary to take into account the relevant controls, and if so, what were their reference values. Describe the stimulation conditions or an annotate that all steps have been performed according to the manufacturer's protocol. We have described the protocol followed for the IGRA test according to manufacturer recommendations.

There is not description of the statistical methods used. We have included a description of the statistical methods used.

There is not information whether the research was approved by the bioethical committee. We have included information about ethical approval and written informed consent of the participants.

The results of the qualitative detection of antibodies against the nucleocapsid protein should be provided. This information has been included, along with number of subjects with evidence of past COVID-19 at time 2, as requested.

On the basis of which criteria were selected people who underwent the IGRA test and how many actually were there should be provided. All residents and workers in one single center that was randomly selected were assessed for cellular immunity at time 2. There were 5 subjects that had negative results in the IGRA. We have indicated that the small number of subjects tested for IGRA weakens any conclusion about cellular immunity in our study. All, this information has been provided in the text.

The health condition of the participants themselves (this characteristic is missing) and the existing diseases are probably not without significance, and they will rather affect the humoral response, not the age itself, in the compared groups. We do not have this information. We have included this caveat as a limitation of the study.

Data (if available) should be included concerning reinfection or SARS-CoV-2 infection following full vaccination in these groups. We do not have this information.

There were no detailed analyzes of the influence of age itself (the inference is rather indirect, based on the assumption that the charges are older than the staff). A linear regression analysis has been included to analyze the independent contribution of age, gender, personal status, and history of COVID-19 over IgG-S titers after vaccination.

Language and style has been reviewed by an native-English medical writer.

Reviewer 2 Report

 The purpose of this study was to investigate the humoral and cellular response in nursing home residents and staff after vaccination with Pfizer's vaccine. The study included a sufficient number of participants (either
residents or staff). Some questions that need clarification:

 Major:
1. Please explain how the statistical analysis (all statistical tests used, software) was done. Nothing is mentioned
in the current form. The description of the methods should be in the Method section and in some places when the corresponding results are presented.
Figure 2: I have some doubts about the true relationship between IFN activity and IgG titers.
- What is this line? It does not seem to be the regression line, because on visual inspection it does not seem to
be the line that best fits the data.
- Left panel: there is a discernible curvilinear relationship leading to a plateau, not a linear one
- Right panel: the linearity does not seem to be true, as the point at (2.0, -4.0) is an outlier. If you omit this point, the slope of the regression line would change dramatically. It would even be close to zero, indicating that there is no relationship between IFN A2 activity and IgG-S titers.
Please comment on this in your manuscript and modify accordingly.

Minor:
1. Abstract: Please clarify what the numbers in parentheses refer to.

Author Response

For completeness, the authors could add a section on the clinical aspects of rotavirus infection (symptoms, transmission, diagnosis, complications, and common treatment measures available today). We feel this comment does not apply to our paper.

Round 2

Reviewer 1 Report

The comments that I made to the previous version of the work have been fully taken into account.

Author Response

These are the responses to Reviewer 2 comments

  1. Please explain how the statistical analysis (all statistical tests used, software) was done. This information has now been included in Methods.
    2. Figure 2: I have some doubts about the true relationship between IFN activity and IgG titers. What is this line? This is the trend line that is provided by the statistical software. The correlation analysis was done using Spearman´s Rho test. We have included a comment about the weakness of the correlation as suggested.

Minor:
1. Abstract: Please clarify what the numbers in parentheses refer to. This has been done.

Reviewer 2 Report

My previous comments have been addressed